# Quality of Life in Amazonian Women during Cervical Cancer Treatment: The Moderating Role of Spirituality

**DOI:** 10.3390/ijerph20032487

**Published:** 2023-01-30

**Authors:** Rosana Moysés, Inês Marques, B. Daiana Santos, Adele Benzaken, M. Graça Pereira

**Affiliations:** 1Psychology Research Centre, School of Psychology, University of Minho, 4710-057 Braga, Portugal; 2School of Medicine, Federal University of Amazonas, Manaus 69067-005, Brazil; 3Oswaldo Cruz Foundation, Leônidas and Maria Deane Institute, Manaus 69057-070, Brazil

**Keywords:** cervical cancer, spirituality, psychological morbidity, illness perception, quality of life

## Abstract

This study aimed to examine the contribution of psychological variables to quality of life (QoL) of Amazonian women and to analyze the moderating role of spirituality in the relationship between psychological morbidity and Qol and between illness perception and QoL. This cross-sectional study included 119 women undergoing treatment for cervical cancer (CC). The Pearson correlation test was used to evaluate the relationship between sociodemographic, clinical, and psychological variables. To test how psychological morbidity, illness perception, and spirituality contribute to QoL, a path analysis was performed and to test the moreating role of spirituality, a moderation analysis was conducted. The results revealed that the presence of symptoms, high psychological morbidity, negative body image, and threatening illness perception were predictors of lower QoL. Spirituality moderated the relationship between psychological morbidity and QoL, and between illness perception and QoL. The moderating role of spirituality emphasizes its role as a coping strategy and should be included in cancer treatment. Interventions should target psychological morbidity, threatening illness perception, and address women’s concerns with body image and sexual concerns. CC treatment should include interprofessional healthcare teams addressing the biological and psychosocial factors of Amazonian women. As a result of this study a mobile application to monitor women’s health, adapted to cultural and social characteristics, was created.

## 1. Introduction

The American Cancer Society reported the emergence of about 14,480 new Cervical Cancer (CC) cases in the United States, which accounted for approximately 4290 deaths in 2021 [1]. In Brazil, in 2020, 16,590 new CC cases emerged, affecting about 15.43 per 100,000 women. Given these figures, CC is the third most common type of cancer in the country and the second most common cancer in women [2].

Amazonas is the Brazilian state with the largest territorial size, located in the heart of the Amazon Rainforest. It is also the largest river basin in the world [3]. The life of the Amazonian population is intrinsically linked to the rivers and the forest. Travel is mainly by river, a fact that defines access to services, education, and health [4]. The population of the state of Amazonas is 4.081 million inhabitants, and it has the largest indigenous population in Brazil, which includes more than 50 ethnicities [5]. The state has 62 municipalities, but the largest concentration of people is in the city of Manaus, the capital of the state, with just over 2 million inhabitants. All of the specialized health services are located in this city, which, for many, results in the need to travel for hours and days, generally by the river, from other municipalities to receive appropriate health care services [4].

In 2019, in the state of Amazonas, 580 new cases of CC were predicted for the following year. Based on these predictions, Amazonas is the state in the northern region of Brazil with the second highest number of cases and mortality rate from CC [2]. These statistics may be explained by the difficulty many women have in accessing preventive gynecological examinations (Pap smear) and diagnosis of advanced stages of the disease. 

Cervical cancer is a chronic disease with anatomical-physiological, psychological, and social consequences that impact patients’ lives from the time of diagnosis and throughout the treatment, depending on several factors in the trajectory of the illness and treatment [6,7].

Several studies have addressed the importance of clinical variables, such as age at diagnosis, type of treatment, and cancer stage, on women’s quality of life (QoL). The average age for a CC diagnosis is 50 and often diagnosed between the ages of 35 and 44, being a risk factor for CC, although HPV (human papillomavirus) infection increased the chances of developing CC at any age [2]. Effective therapy for CC may result in a cure rate of 80% in early-stage disease, stages I–II, and of 60% in stage III [1,8].

The treatment for CC (brachytherapy or chemotherapy), whether in the early or advanced stages, may cause physical side effects such as low levels of lubrication and vaginal stenosis (related to brachytherapy), hair loss, gastrointestinal problems, changes in sense of taste, loss of appetite, urinary incontinence, skin lesions, lymphedema, insomnia, pain during sex, and hormonal and reproductive function disorders impacting the patient’s quality of life [9]. 

A diagnosis of CC is associated with psychological morbidity. A woman with CC experiences anxiety and depression when diagnosed, and four to six months later, these symptoms persist [10]. Psychological morbidity in patients with CC is also associated with illness perception. Patients with gynecological cancer report more threatening illness perceptions that are associated with greater psychological distress and a worse QoL [11].

According to research, psychological morbidity (anxiety and depression) occurs in two-thirds of cancer patients, and the main risk factors for the development of anxiety in cancer patients are associated with a history of trauma and anxiety [12]. Monitoring psychological morbidity (anxiety and depression) in patients with cancer is extremely relevant, as studies showed that these psychiatric disorders are associated with a decreased adherence to treatment, an increase in duration and number of hospitalizations, a worse social functioning that impacts QoL, and an increase in patients’ mortality rate [12,13]. Patients with gynecological cancers who are most at risk of developing symptoms of psychological morbidity are those with low education and less knowledge regarding the disease and its treatment, particularly in Latin America [13]. 

When dealing with adversity such as the diagnosis of CCU, spirituality may function as a coping strategy that provides new meanings and helps the patient to reorganize his/her cancer experience and all the adaptation tasks it involves. Several studies in cancer patients have described a strengthening of spiritual perceptions, perhaps as a result of a reflection regarding one’s finite existence of and the uncertainties caused by the disease [14,15].

In Latin cultures, such as the Brazilian, spirituality is an everyday aspect of an individual’s life, and therefore, the way of dealing with cancer is directly influenced by patients’ spiritual beliefs [14]. For women with CCU, spirituality may be a coping strategy that provides family and social support networks. The literature recognizes the non-unanimity of the concepts of religiosity and spirituality, [15] and in Brazil, women with CC have difficulty in differentiating between spirituality and religiosity since, in most studies, many women consider the two terms similar [15,16]. Whether at diagnosis or during treatment, spirituality is described, by patients, as a support strategy that helps them accept the disease [16].

Several studies showed evidence of an association between quality of life and religiosity/spirituality in a wide distribution of regions and cultures around the world. Intervention studies focused on spirituality also showed good health outcomes, and an association between spirituality and QoL, particularly when the intervention includes spiritual counseling or contemplative meditation across different chronic diseases [15,16]. Studies on spirituality in cancer patients reveal that spirituality is an important strategy for coping with the disease, associated with survival and support for life reorganization, helping to re-signify the events imposed by cancer [16,17]. The literature also shows that religious/spiritual involvement is associated with greater physical well-being, improved QoL, and a more efficient immune system. In patients with chronic diseases, such as cancer, spiritual well-being has been associated with lower rates of psychological morbidity [17,18]. 

Illness perception in women with CC may be influenced by spirituality since the latter may provide meaning and enhances positive beliefs of faith and hope associated with less threatening perception and psychological distress [19]. Therefore, it becomes important to assess whether spirituality may moderate the relationships between psychological morbidity and QoL, as well as between illness perception and QoL. 

Amazonian women have to travel for days to receive a CC diagnosis and treatment, having also to be separated from their families during CC treatment. There is a gap in the literature concerning the psychological impact of CC on QoL, in this particular population, including the contribution of spirituality, despite the high incidence of CC in the state. This is the first study that considers sociodemographic, clinical, and psychological variables as predictors of Qol, in this population. Therefore, the relevance of the present study is to contribute to a better understanding of psychological morbidity, illness perception, and spirituality regarding QoL in Amazonian women. We hope to contribute also for the design of interventions that promote QoL in this population that may also be culturally relevant for other Latin American countries. 

## 2. Materials and Methods

### 2.1. Study Design

This study used a cross-sectional design. Data collection took place in the reference hospital for CC treatment in the state of Amazonas. 

### 2.2. Participants

The sample was a non-probabilistic convenient sample, calculated using the software Sample Size Calculator by Raosoft based on the estimated number of new cases of cervical cancer treated at FCECON in 2016 (170 women), considering a significance level of 5% with a 95% confidence interval, and requiring a minimum sample of 119 patients.

The study included 119 women, over 18 years of age, from the state of Amazonas, who had been diagnosed with CC and were undergoing surgery and/or chemotherapy/radiotherapy. Exclusion criteria included the following: being of an indigenous background due to cultural and linguistic peculiarities and the presence of severe psychiatric disorders, such as schizophrenia, schizotypal disorders, and delusional disorders registered in the patient’s medical records. Randomization and recruitment of patients occurred between August 2017 and October 2018. 

This study was carried out in the only public hospital for the treatment of oncology diseases in the state of Amazonas, founded in 1974, that aims to treat and reduce the incidence and mortality from cancer, in the state [19].

### 2.3. Instruments

Sociodemographic and Clinical Questionnaire: This instrument evaluated sociodemographic and clinical variables such as age, education, cancer stage, age at diagnosis, and type of treatment.

Cancer QoL Core 30 Questionnaire (QLQ-C30): [20,21]. This instrument has 30 items and includes 5 functional scales (physical, functional, emotional, social, and cognitive), a scale on the global health status with 3 symptom scales (fatigue, pain, and nausea), and 6 additional symptom items (dyspnea, insomnia, loss of appetite, constipation, diarrhea, and financial difficulties). The original version showed a Cronbach’s alpha of 0.70 for the total scale. The Brazil version has the same number of items as the original version and Cronbach’s alphas ranging between 0.72 and 0.86. In the present study, Cronbach’s alpha was 0.89 for the total score. For the additional symptom’ scales, higher scores indicated poor QoL, and in the functional and global health status scales, higher scores indicate a better QoL.

Cervical Cancer Module (QLQ-CX24) [22,23]: This instrument is the specific module for CC and has 24 questions, which are divided into 3 multi-item scales, and address symptoms experience (composed of 11 items), body image (composed of 3 items), and sexual/vaginal function (composed of 4 items), and 6 single-item scales, lymphedema, peripheral neuropathy, menopausal symptoms, sexual concerns, sexual activity, and sexual pleasure. Patients have the option of not responding to the questions on the scale of sexual/vaginal function and sexual pleasure when they are not sexually active. The Cronbach’s alpha of the original version was 0.70. The Brazilian version has the same number of items and Cronbach’s alphas ranging between 0.75 and 0.77. In this study, Cronbach’s alpha was 0.68 for the symptom experience subscale, 0.78 for the body image subscale, and 0.54 for the sexual/vaginal functioning subscale. The latter subscale was not used because the response rate was very low (*n* = 5). In the subscales, symptom experience, lymphedema, peripheral neuropathy, and menopausal, a high result indicates more symptoms, and in the body image subscale, a high result indicates a worse perception of body image.

Hospital Anxiety and Depression Scale (HADS) [24,25]: This instrument includes 14 items scored from 0 to 3, resulting in a maximum score of 21 for each subscale (anxiety and depression); the higher the score, the greater the number of symptoms of anxiety and depression. A global score indicates psychological morbidity or distress. The Cronbach’s alpha of the Brazilian version was 0.68 for the anxiety subscale and 0.77 for the depression subscale. The authors did not report the alpha for the global scale. The Cronbach’s alpha in the present study was 0.71 for the anxiety subscale, 0.72 for the depression subscale, and 0.82 for the global scale. In this study, only the full scale was used.

Revised Brief Illness Perception Questionnaire (Brief IPQ-R) [26,27]: This instrument is a reduced version of the Illness Perception Questionnaire (IPQ-R) [28]. The instrument has 9 items assessing the cognitive perception, emotional perception, and understanding of the disease, with 8 items scored on a 10-point Likert scale. In the original version, Cronbach’s alpha ranged between 0.75 and 0.89. In the Brazilian version, the global scale was used, and Cronbach’s alpha was 0.64. In this study, Cronbach’s alpha was 0.60. High scores indicate a more threatening perception of the disease.

Spiritual and Religion Attitudes in Dealing with Illness Questionnaire (SPREUK) [29,30]: The instrument assesses how patients with chronic diseases use spiritual and religious coping. The instrument includes 15 items grouped into 3 subscales: the seeking of support/access; trust in a superior guidance/source; and reflection and positive interpretation of the disease. The original version showed Cronbach’s alpha ranging from 0.86 to 0.91 and 0.94 for the full scale. Higher scores indicate more use of spirituality. The Brazilian validation of the instrument has one less item and the items were grouped into two domains: positive interpretation of the disease and spiritual support. Confirmatory factor analysis showed a good model fit (Chi-square = 93.87, CFI = 0.951, TLI = 0.94, RMSEA = 0.045, RMSEA upper 90% CI = 0.072). The Cronbach’s alpha for the spiritual support subscale was 0.82 (9 items) and, for the reflection and positive interpretation of the disease subscale, 0.81 (5 items), and for the full scale, a Cronbach’s alpha of 0.87. Higher scores indicate greater use of spirituality.

### 2.4. Procedures

The study was conducted at the FCECON Chemotherapy and Radiotherapy Outpatient Clinic, and the patients were identified by nurses and invited to participate in the study. Patient participation was voluntary, with all participants signing an informed consent form. Given the high illiteracy rate in the Amazon, the questionnaires were applied using a face-to-face interview format, and, for illiterate people, the consent form was read to them, and, when they agreed to participate, their fingerprint was taken. After signing the informed consent, the patients answered the questionnaires. Clinical data were collected from the patient’s medical records. Patients were randomly selected on alternate days of the week during radiotherapy/chemotherapy sessions.

### 2.5. Data Analysis

Data were analyzed using the IBM SPSS^®^ (Statistical Package for the Social Sciences) program, version 26.0. Descriptive statistics were used to describe the sociodemographic and clinical characteristics of the sample. When the assumptions for the use of parametric tests were fulfilled, Pearson’s correlation test was used to evaluate the relationship between sociodemographic, clinical, and psychological variables. 

To test how psychological morbidity, illness perception, and spirituality contribute to QoL, a path analysis was performed. Only the variables that correlated with Qol were entered into the model. To determine the modifications of the adjusted model, chi-square, TLI, and RMSEA indexes were used following the appropriate reference values. 

The moderating role of spirituality in the relationship between psychological morbidity, illness perception, and QoL was assessed using the Macro-Process command in the SPSS program and the Johnson–Neyman (JN) technique, since all the assumptions for moderation were fulfilled. Since the Brazilian version of Spreuk is different than the original version, the two subscales were used.

## 3. Results

### 3.1. Sample Description

The sample consisted of 119 Amazonian women with CC, aged between 26 and 75 years of age (M:47.54, SD: 11.691), with most of the patients coming from the interior of the state of Amazonas (53.8%). Of the total sample, 36.1% were in a common law relationship and 34.5% were married. Concerning education, most women had completed secondary education (31.9%). Regarding employment status, the majority of women did not work outside the home (31.8%) and had a low socioeconomic status. Of the total sample, the majority (96%) claimed to have a religion and only 4% reported having no religion. Table 1 describes the sociodemographic and clinical characterization of the sample.

Regarding psychological characteristics, based on instrument metrics, participants in this study reported more symptoms of depression than anxiety, high use of spirituality, and moderate quality of life. Table 2 describes the psychological characteristics of the sample. 

### 3.2. Relationships between All Variables: Path Analysis

Before the performance of the path analysis, correlations between all variables and QoL were performed. No relationship was found between clinical and psychological variables with QoL. A negative association between illness perception (*r* = −0.479, *p* < 0.01), psychological morbidity (*r* = −743, *p* < 0.01), body image (*r* = −0.469, *p* < 0.01), specific symptoms/symptom experience (*r* = 0.614, *p* <0.01), and QoL was found. Moreover, the presence of lymphedema (*r* = −0.243, *p* < 0.01), peripheral neuropathy (*r* = −0.291, *p* < 0.01), menopausal symptoms (*r* = −0.409, *p* < 0.01), and sexual concerns (*r* = −195, *p* < 0.05) were negatively correlated with QoL. The path analysis showed a good fit with the data: Chi-square = 1.22; df = 2; TLI = 0.991; CFI = 0.998; RMSEA = 0.044 = <0.01. Disease symptoms negatively predicted QoL (β = −0.27). The symptoms of CC treatment positively predicted psychological morbidity (b = 0.35) and psychological morbidity negatively contributed to QoL (β = −0.57). Body image positively predicted psychological morbidity (β = 0.31), and psychological morbidity negatively predicted QoL (β = −0.57). Illness perception positively predicted psychological morbidity (β = 0.34) and psychological morbidity negatively contributed to QoL (β = −0.57). Based on the correlations, Figure 1 describes the initial model that was tested, and Figure 2 the final model.

### 3.3. Spirituality as a Moderator between Psychological Morbidity and QoL

The model that tested the moderating role of spirituality (reflection and positive interpretation of disease subscale) in the relationship between the patient’s QoL and psychological morbidity showed significant results, F (3.115) = 18.88, *p* < 0.001, b = 0.021, 95% CI [−1.5, −0.62], t = 0.02, *p* < 0.05, and explained 33% of the variance. There was a negative relationship between psychological morbidity and QoL when the use of spirituality was higher (β = −1.07, 95% CI [−1.5297, −0.6291], t = −4.74, *p* < 0.001). The JN technique revealed that psychological morbidity correlated negatively with QoL when the standardized value of spirituality was 118.27 below the mean (β = −0.6871, *p* = 0.05), and this was true for 99% of the sample (Figure 3). Therefore, the negative relationship between psychological morbidity and QoL happened in patients who used more spirituality as a strategy.

Spiritual support (subscale) did not moderate the relationship between psychological morbidity and QoL (β =1.13, 95% CI [0.9365, 0.0343], t = −0.0798, *p* > 0.05). 

### 3.4. Spirituality as a Moderator between Illness Perception and QoL

When testing the model, the moderating role of reflection and positive interpretation of disease (subscale) between illness perception and patients’ QoL was significant, F (3.115) = 6.63, *p* < 0.001, b = 0.13, 95% CI [0.002;028], t = 0.02, *p* < 0.05, which explained 14.7% of the variance. There was a negative relationship between illness perception and QoL when spirituality was lower (β = −0.34, 95% CI [−0.6866, −0.0121], t = −2.05, *p* < 0.001). The JN technique showed that illness perception correlated negatively with QoL when the standardized value of spirituality was 100.55 below the mean (β = −0.3493; *p* = 0.05), which occurred in 98% of the sample (Figure 3). Therefore, the negative relationship between illness perception (threatening) and QoL happened in patients who used less spirituality as a strategy. 

Spiritual support (subscale) did not moderate the relationship between illness perception and QoL (β = 0.65;95% CI [0.5426, 0.0282], t = −0.61, *p* > 0.05).

## 4. Discussion

The literature describes the importance of spirituality in patient care and QoL [15]. In the case of patients from Latin American cultures, such as Brazil, spirituality is considered an everyday event. Therefore, the way patients deal with cancer has a direct influence on their spiritual beliefs and religious practice [14,15]. In this perspective, spirituality is an important instrument for a patient’s recovery since it provides support and helps manage the difficulties, as well as being a reason for gratitude and encouragement based on one’s faith [14]. 

Studies involving cancer patients in Brazil have shown that patients with CC use spirituality and religiosity as coping mechanisms to deal with suffering since it provides these patients with the ability to develop resilience, hope, and faith. Brazil has the second highest number of Christians in the world; [3] the population is 60.6% Catholic and 28.5% evangelical/protestant. Several studies have shown the relevance of spirituality/religiosity on illness in the population [31]. In the northern region, where the state of Amazonas is located, the predominant religion is evangelical [3].

Some studies found that patients with CC and lower psychological morbidity (anxiety and depression) reported a better QoL, and among the factors that help minimize the symptoms, spirituality is considered an important strategy, which confirms the results of the present study [14,32,33]. The fact that patients have a religion is associated with greater spiritual well-being, greater acceptance of the disease, and better emotional control [17,32]. 

The results also showed a negative association between psychological morbidity and QoL. Several studies have shown the presence of high levels of psychological morbidity in women with CC, both at the time of diagnosis and during treatment. The diagnosis and treatment have several consequences at social, physical, sexual, and psychological levels. These psychological consequences of diagnosis and treatment for CC consequently lead to a reduction in patients’ QoL [33,34].

In addition, a negative association between illness perception and QoL was found. Studies have shown that the diagnosis of cancer is associated with a low perception of QoL even before treatment [34,35]. Illness perception also has an impact on how the individual perceives the symptoms, the duration of the disease, and the time needed for recovery, which may also impact QoL [35].

Several cancer treatments have side effects that may cause psychological and bodily changes in women. Changes in appearance and functional changes can thus lead to dissatisfaction with body image. The literature reveals that women with cancer undergoing treatments that affect body parts associated with femininity and sexuality are more vulnerable to a change in their perception of body image, thus compromising their QoL [34,36].

Studies have shown that patients undergoing CC treatments report more symptoms, such as fatigue, nausea, pain, insomnia, loss of appetite, and diarrhea, which result in a negative impact on their QoL. The symptoms experienced by the patients may also contribute to the development of psychological morbidity, compromising QoL. Studies have revealed that, during treatment, there are increased sexual concerns [37]. In fact, according to Abbott-Anderson and Kwekkeboom [36], CC treatment may have negative consequences on sexual, social, and psychological functions, thus compromising QoL. Several studies have also described the relationship between side effects and QoL following the diagnosis of gynecological cancer and its treatment [38].

In this study, psychological morbidity predicted Qol as expected. After the diagnosis of CC, several changes occur in women’s lives that may be experienced as negative, such as psychological morbidity, that may remain throughout the treatment [39,40].

Caring for patients with CC should consider strategies based on spirituality and provide psychological support to promote treatment adherence and address illness perception [15,33,34]. Multiprofessional oncology teams in Brazil recognize the relevance of spirituality in the care of cancer patients, but also recognize that healthcare professionals need to be prepared to make use of spiritual beliefs as an instrument to strengthen the monitoring of cancer patients during the progression of the illness [40].

In the present study, spirituality moderated the relationship between psychological morbidity and QoL, i.e., patients reported higher use of spirituality when there is a negative relationship between psychological morbidity and QoL. Spirituality may be used as a buffer against the negative impact of psychological morbidity on Qol. Other studies have also shown an association between spirituality and lower psychological morbidity (depression and anxiety), and better QoL in patients under treatment [33].

Spirituality also moderated the relationship between illness perception and QoL. When patients showed a lower use of spirituality, there is a negative relationship between threatening illness perception and QoL. This result is in agreement with the literature, which shows that spirituality may indeed work as a coping strategy, contributing to a better QoL, [41] since it helps the patient to give meaning and better accept the disease. Spiritual support was not a moderator in the relationship between psychological morbidity and QOL, and between illness perception and QoL, probably because, in Brazil, there are spiritual support groups for patients with chronic diseases in all churches (particularly in those the participants reported to belong) and hospitals, and 96% of the sample reported being religious. Therefore, the sample may have been too homogenous since spiritual support is not an issue, and, therefore, had no impact on QoL [41]. Future studies should assess the moderating role of spiritual support in samples where church spiritual support is not available.

## 5. Conclusions

This study found that more symptoms of anxiety and depression and a more threatening illness perception contributed to a decreased QoL and worse body image, and more sexual, physical, and menopausal symptoms also negatively impacted QoL. In addition, spirituality moderated the relationship between illness perception and QoL, and between psychological morbidity (anxiety and depression) and QoL.

Based on the results, interventions should target psychological morbidity and threatening illness perception that predict lower QoL and also addresses women’s concerns with body image and sexual concerns. Since spirituality played a moderating role between psychological morbidity and QoL, and between illness perception and QoL, it is important to include this dimension (coping strategy) in cancer treatment as an inner resource to promote QoL, in this population.

Finally, in Amazonian women, care in CC treatment should include interprofessional healthcare teams addressing both the biological and psychosocial aspects of CC. Future studies should also focus on the contribution of partners and caregivers psychological variables to women’s QoL.

## 6. Implications

Spirituality played an important role as a relevant strategy to face CC in Amazonian women. The QoL of patients, as reported in the literature, is influenced by psychosocial variables, but in the Amazonian perspective, spirituality has an expressive buffering role regarding the symptoms of anxiety and depression, as well as on illness perception.

Based on the results of the present study and on the epidemiological profile of CC in Amazonian women, in a supportive home for women undergoing cancer treatment in the city of Manaus, spirituality is now being included in health promotion workshops, providing therapeutic moments of discussion about the cancer experience. Moreover, considering the results, a mobile application (ELLA) was created and widely disseminated (https://www.youtube.com/watch?v=RIx9m5JDizc&list=PLrfYYM4qqwLOGkCfFG1uyJCqpAJDLThXV, accessed on 23 November 2022) that includes psychoeducation regarding CC to help women record exams, menstrual cycles, and information about the use of contraceptives, in a simple and easy way by using voice commands. This mobile application has become available for all Brazilian women and may be used by Portuguese-speaking women worldwide as well.

## Figures and Tables

**Figure 1 ijerph-20-02487-f001:**
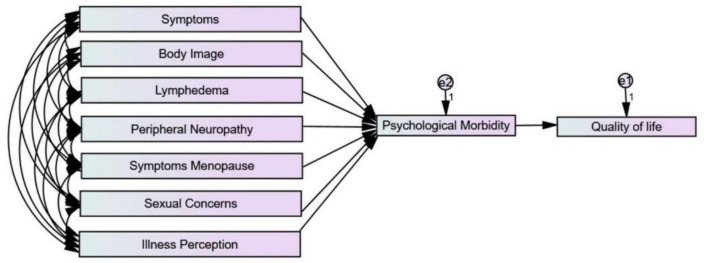
Initial model tested.

**Figure 2 ijerph-20-02487-f002:**
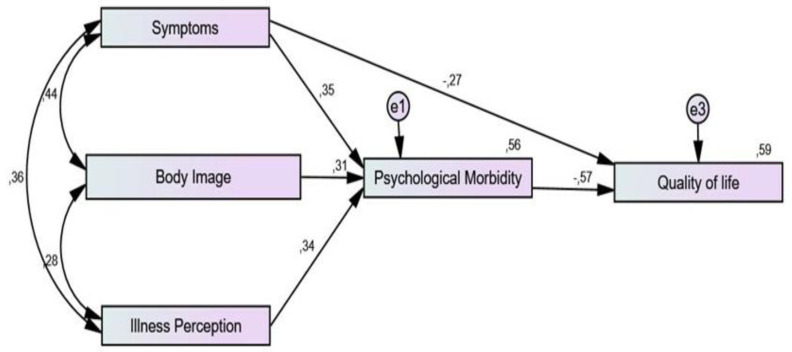
Final model: chi-square = 1.22; df = 2; Tucker–Lewis index = 0.991; comparative fit index = 0.998; root mean square error of approximation = 0.044.

**Figure 3 ijerph-20-02487-f003:**
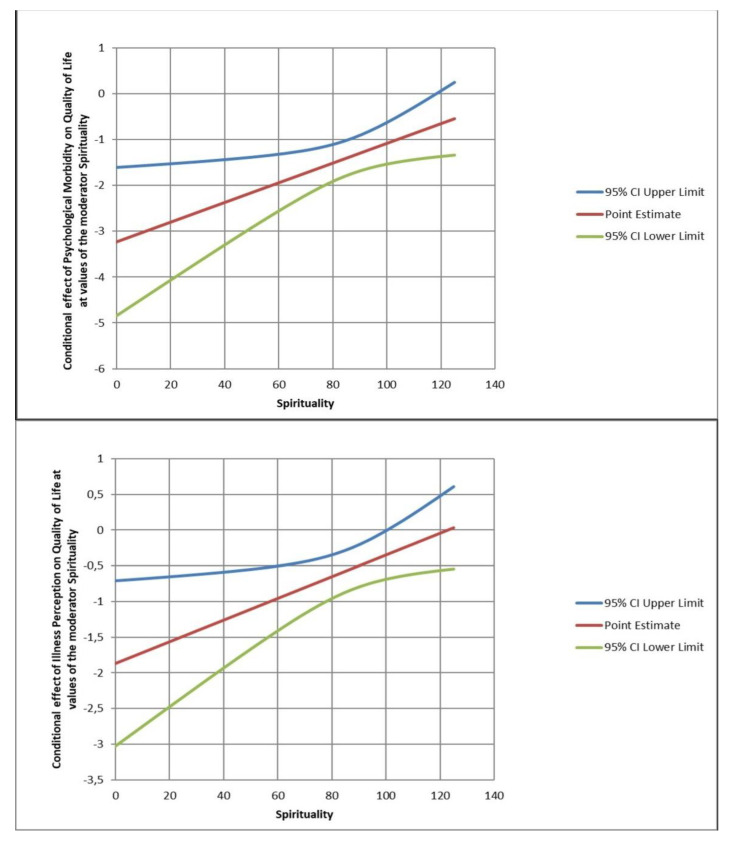
The moderating effect of spirituality (reflection and positive interpretation of disease and spiritual support) in the relationship between psychological morbidity and QoL and between illness perception and quality of life.

**Table 1 ijerph-20-02487-t001:** Sociodemographic and clinical characteristics of the sample group.

	*n* (%)	M (SD)	Min	Max
Age	119	47.54	26	75
Age at Diagnosis	119	46.69 (11.95)	26	77
Education				
<1 Year	32 (26.9)			
1 to 3 Years	24 (20.2)			
4 to 7 Years	21 (17.6)			
8 to 10 Years	42 (35.3)			
Religion				
Evangelical/Protestant	72 (63)			
Catholic	34 (29)			
Spiritism (Kardecist)	8 (4)			
Stage				
Stage I	15 (12.6)			
Stage II	41 (34.5)			
StageIII	38 (31.8)			
Stage IV	25 (21.1)			
Treatment				
Chemotherapy	3 (2.5)			
Radiotherapy	3 (2.5)			
Surgery/Radiotherapy	4 (3.3)			
Surgery/Chemotherapy	2 (1.6)			
Chemotherapy/Radiotherapy	70 (58.8)			
Surgery/Radiotherapy/Chemotherapy	37 (31.3)			

**Table 2 ijerph-20-02487-t002:** Psychological characteristics of the sample group.

	*n*	M (SD)	Min	Max
Quality of Life	119	63.73 (19.43)	21	99
Psychological Morbidity	119	14.51 (7.67)	1.0	37.0
Anxiety		6.95 (4.30)	0	19
Depression		7.63 (4.35)	0	19
Spirituality	119	90.97 (11.03)	52	125
Reflection and Positive Interpretation		90.28 (8.53)	0	125
Spiritual Support		83.74 (18.26)	48	113
Illness Perception	119	37.80 (11.35)	9	63
M (Mean) DP (Standard Deviation)				

## Data Availability

The data presented in this study are available on request from the corresponding author.

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
