# Peer review of "Quality of Life in Amazonian Women during Cervical Cancer Treatment: The Moderating Role of Spirituality"

_ijerph, 2023, doi:10.3390/ijerph20032487_

Round 1

Reviewer 1 Report

The topic is interesting and very much relevant to the current context, moreover, the review paper has wider theoretical and practical applications. The authors have put their best efforts to execute this paper. However, I have the following reservations and suggestions for the sake of improvement of the undertaken study:

1) The logical sequence of the abstract should be as 1) objectives, 2) methodology, 3) Findings, 4) conclusion and 5) implications. Thus, the authors should also rewrite the abstract in this sequence. The authors should mention the review analysis method in the methodology, after findings conclusion and then please describe important implications. 

2) The authors did not establish the motivation, significance, and novelty of the undertaken study. The authors are suggested to improve this important factor in the "Introduction" section. The background of research should also be presented in the section. The structure of the review paper should also be presented in the end of Introduction section.

3) The conclusion should be added after the discussions section, conclusion is always one step ahead of findings. 

4) The practical, theoretical and societal implications should be discussed after the conclusion, and in the light of the conclusion and discussions. 

5) Minor spelling and grammatical mistakes should be improved.

Author Response

Dear Reviewer

Thank You for the opportunity to revise the paper. Your suggestions greatly improved the paper. All the comments were addressed and to make the reading easier they were placed in blue colour in the manuscript.

_______________________________________________________________

1) The logical sequence of the abstract should be as 1) objectives, 2) methodology, 3) Findings, 4) conclusion and 5) implications.  Thus, the authors should also rewrite the abstract in this sequence. The authors should mention the review analysis method in the methodology, after findings conclusion and then please describe important implications. 

Answer: The abstract was revised (Page 1, lines 9-26).

2) The authors did not establish the motivation, significance, and novelty of the undertaken study. The authors are suggested to improve this important factor in the "Introduction" section. The background of research should also be presented in the section. The structure of the review paper should also be presented in the end of Introduction section.

Answer: The Introduction was revised, explaining in detail the relevance the study (Page 2, lines 75-91; Page 3, lines 101-108; 118-119; 122-123; 126-127).

3) The conclusion should be added after the discussions section, conclusion is always one step ahead of findings.

Answer: The Conclusion is after the discussion section. (Page 11, lines 439-453).

4) The practical, theoretical and societal implications should be discussed after the conclusion, and in the light of the conclusion and discussions. 

Answer: An implication section after the conclusion section was added  (Page 11, lines 454-464; Page 12, lines 465-468).

5) Minor spelling and grammatical mistakes should be improved.

Answer: The manuscript was proofread.

Reviewer 2 Report

The authors discussed the moderating role of spirituality in the improvement of QOL of CC patients. some concerns should be addressed.

1. There is too much unrelated information in the introduction part, especially the several paragraphs at first. This part should be revised thoroughly, information about the effects of spirituality on related diseases should be added.

2.  sample size is particularly important to derive a significant result, sample size estimation is an essential procedure before the conduction of study.

3. The procedures part should be revised, it is confusing to understand how the study was conducted following the descriptive.

Author Response

Dear Reviewer

Thank You for the opportunity to revise the paper. Your suggestions greatly improved the paper.

All the comments were addressed and to make the reading easier they were placed in blue colour in the manuscript.

___________________________________________________________

1)There is too much unrelated information in the introduction part, especially the several paragraphs at first. This part should be revised thoroughly, information about the effects of spirituality on related diseases should be added.

Answer: The Introduction was revised, including spirituality (Page 2, lines 86-91; Page 3, lines 101-108).

2) Sample size is particularly important to derive a significant result; sample size estimation is an essential procedure before the conduction of study.

Answer: The text was revised (Page 3, lines 133- 143)

3) The procedures part should be revised; it is confusing to understand how the study was conducted following the descriptive.

Answer: The procedures was revised (Page 5, lines 207-215).

Reviewer 3 Report

A study is designed to investigate the relationship between quality of life, and The moderating role of spirituality in patients with cervical cancer.  For this reason, it would be helpful to classify the conditions of selected patients according to the following points.

Negative CRE: the level of negative religious/spiritual coping practiced by the individual,

Positive CRE: the level of positive religious/spiritual coping practiced by the individual,

Religious/spiritual coping rate,

Patients' belief that chemotherapy is a good treatment, in addition and patients' belief levels should be classified.

Reference writing should be reviewed for compliance with the rules.

Author Response

Dear Reviewer

Thank You for the opportunity to revise the paper. Your suggestions greatly improved the paper.

All the comments were addressed and to make the reading easier they were placed in blue colour in the manuscript.

____________________________________________________

1) A study is designed to investigate the relationship between quality of life, and The moderating role of spirituality in patients with cervical cancer.  For this reason, it would be helpful to classify the conditions of selected patients according to the following points.

Negative CRE: the level of negative religious/spiritual coping practiced by the individual,

Positive CRE: the level of positive religious/spiritual coping practiced by the individual,

Religious/spiritual coping rate,

Answer: The instrument does not provide positive and negative spiritual coping but information on how patients with a chronic disease use spiritual/religious coping. The three scales are all “positive”: seeking support; trust in a superior guidance; reflection and positive interpretation of the disease. The authors added a table 2 concerning the psychological characterization of the sample (Page 6, line 247-250; Page 7, Table 2).

2) Patients' belief that chemotherapy is a good treatment, in addition and patients' belief levels should be classified.

Answer: The instrument assesses cognitive, emotional and understanding illness perception using a Likert scale with no cut points for level of perceptions.

3) Reference writing should be reviewed for compliance with the rules.

Answer: Bibliographic references were revised following the Journal rules (Page 12, lines 492-516; Page 13, 517-575; Page 14,576-584).

Reviewer 4 Report

The article reported an interesting relationship between spiritual beliefs and the improvement in the quality of life of cancer patients. I recommend accepting this article after considering the following comments

1)    In the introduction, authors are encouraged to elaborate more on the signs of "psychological morbidity."

2)    Why patients having an indigenous background were excluded from the study?

3)    The title of subheading no 2.3 "Instruments" should be replaced by "Methods"

4)    Figure 3 is not clear. Its resolution should be improved.

5)    On several occasions, spirituality has been repeatedly defined as "the reflection and positive interpretation of disease subscale". Such a definition is unnecessary to be repeated and its inclusion once in the article is enough.

6)    The manuscript is poorly written. It is filled with grammatical and formatting errors. It should be thoroughly improved.

Author Response

Dear Reviewer

Thank You for the opportunity to revise the paper. Your suggestions greatly improved the paper.

All the comments were addressed and to make the reading easier they were placed in blue colour in the manuscript.

_____________________________________________________

1)    In the introduction, authors are encouraged to elaborate more on the signs of "psychological morbidity." 80-89

Answer: The Introduction was revised and more information was added regarding psychological morbidity (Page 2, lines 75-85).

2)    Why patients having an indigenous background were excluded from the study?

Answer: The participants section was revised, explaining in detail the exclusion of indigenous patients (Page 3, lines139-142).

3)    The title of subheading no 2.3 "Instruments" should be replaced by "Methods"

Answer: The authors did not understand this suggestion since point 2 is: Materials and Methods and 2.4 really addresses the instruments after 2.1. Study design and 2.2. Participants.

4)    Figure 3 is not clear. Its resolution should be improved.

Answer: The figure was revised.

5)    On several occasions, spirituality has been repeatedly defined as "the reflection and positive interpretation of disease subscale". Such a definition is unnecessary to be repeated and its inclusion once in the article is enough.

Answer: The text was revised.

Round 2

Reviewer 2 Report

none